# A Fast and Efficient Ensemble Transfer Entropy and Applications in Neural Signals

**DOI:** 10.3390/e24081118

**Published:** 2022-08-13

**Authors:** Junyao Zhu, Mingming Chen, Junfeng Lu, Kun Zhao, Enze Cui, Zhiheng Zhang, Hong Wan

**Affiliations:** 1School of Electrical Engineering, Zhengzhou University, Zhengzhou 450001, China; 2Henan Key Laboratory of Brain Science and Brain-Computer Interface Technology, Zhengzhou 450001, China; 3School of Intelligent Engineering, Zhengzhou University of Aeronautics, Zhengzhou 450001, China

**Keywords:** ensemble transfer entropy (*TE_ensemble_*), ensemble local transfer entropy (*te_ensemble_*), dynamic interaction, statistical test method

## Abstract

The ensemble transfer entropy (${TE}_{ensemble}$) refers to the transfer entropy estimated from an ensemble of realizations. Due to its time-resolved analysis, it is adapted to analyze the dynamic interaction between brain regions. However, in the traditional ${TE}_{ensemble}$, multiple sets of surrogate data should be used to construct the null hypothesis distribution, which dramatically increases the computational complexity. To reduce the computational cost, a fast, efficient ${TE}_{ensemble}$ with a simple statistical test method is proposed here, in which just one set of surrogate data is involved. To validate the improved efficiency, the simulated neural signals are used to compare the characteristics of the novel ${TE}_{ensemble}$ with those of the traditional ${TE}_{ensemble}$. The results show that the time consumption is reduced by two or three magnitudes in the novel ${TE}_{ensemble}$. Importantly, the proposed ${TE}_{ensemble}$ could accurately track the dynamic interaction process and detect the strength and the direction of interaction robustly even in the presence of moderate noises. The novel ${TE}_{ensemble}$ reaches its steady state with the increased samples, which is slower than the traditional method. Furthermore, the effectiveness of the novel ${TE}_{ensemble}$ was verified in the actual neural signals. Accordingly, the ${TE}_{ensemble}$ proposed in this work may provide a suitable way to investigate the dynamic interactions between brain regions.

## 1. Introduction

The brain is a highly complex system [1,2]. Multiple interconnected brain regions with specific information processing capabilities interact to support the cognitive tasks [3,4], and the strength and the direction of interactions change dynamically [5,6]. For example, the dynamic interactions between the hippocampus (Hp) and posterior parietal cortex (PPC) have been detected in mental arithmetic tasks [5]. The strength of the information flow from the Hp to the dorsal PPC reaches the maximum during mental arithmetic. The maximum value from the Hp to the central PPC is found in verbal memory recall. In rodent spatial associative tasks, the information flows from Hp to the prefrontal cortex (PFC), but the direction reverses in the sampling period [6]. Therefore, a complete description of these interactions, in terms of both strength and directionality, is necessary to reveal the function and the cooperative work of brain regions.

As a measurement of the information interaction between two signals, transfer entropy (TE) is model-free and does not assume any signal or interaction structure [7,8]. Therefore, it has been widely used in neuroscience [9,10,11,12]. However, TE is the average of information transfer over time. Th application of a sliding window is the most common way to explore the dynamic interaction process within and between brain regions. The neural signals should be divided into continuous (non)overlapping segments [13] or be separated into different epochs according to the task [14,15]; then, TE for each segment (epoch) is calculated. To ensure enough samples for TE estimation, the choice of window length is usually a compromise between the estimation precision and the temporal resolution of the dynamic process. The larger the time window, the lower the resolution.

To improve the temporal resolution, a technique (called the ensemble method) takes advantage of multiple realizations of the dynamic process [16], such as numerous recordings of evoked or event-related potentials/fields [17,18]. By estimating TE from the ensemble members instead of individual trials [19], it allows for a time-resolved analysis of the interaction dynamics. Based on this technique, Gómez-Herrero and his colleagues proposed a data-efficient estimator of probability densities to calculate TE from an ensemble of realizations [20]. Additionally, Wollstadt [21] combined the ensemble method with the TE self-prediction optimality (${TE}_{SPO}$) estimator, which was introduced by Wibral [22] to achieve the optimal estimation of delayed information transfer in an ensemble of independent repetition trials. In the following, we use the ensemble transfer entropy $\left({TE}_{ensemble}\right)$ to indicate the transfer entropy estimated from an ensemble of realizations with the ${TE}_{SPO}$ estimator. Instead of TE estimation for each trial, a single ${TE}_{ensemble}$ metric value can be accurately estimated from the ensemble members. ${TE}_{ensemble}$ is not only suitable for short-time data but also for non-stationary signals, which are commonly observed in neuroscience. So, ${TE}_{ensemble}$ can be used for analyzing the dynamic interaction processes between neural signals.

However, the ${TE}_{ensemble}$ algorithm is still at the theoretical level and cannot be applied to the actual neural signals due to its enormous amount of calculation [21]. Firstly, to estimate the ${TE}_{ensemble}$ metric value accurately, a mutual information estimation method proposed by Kraskov, Stogbauer, and Grassberger (KSG) is used [23]. The KSG estimator spends most of the CPU time searching for neighbors, especially in high-dimensional spaces. The complexity of this algorithm is $O(\left(n\times N{)}^{2}\right)$, where n and N are the number of independent repetitions and the sample size in a trial, respectively. The computational complexity is much larger than the methods based on partitioning the observation space (the complexity is O(n×N)) [24]. Secondly, constructing the null hypothesis distribution in ${TE}_{ensemble}$ leads to the calculation increasing several orders of magnitude. In TE, the null hypothesis distribution can be constructed by one set of surrogate data (n trials) [9]. However, only a single metric value can be obtained from n trials in the ensemble method, so multiple sets (usually more than 500) of surrogate data are needed to construct the null hypothesis distribution [21]. The computational complexity of TE with KSG estimator is $O\left(2n\times N^{2}\right)$, but for ${TE}_{ensemble}$, the complexity is $O\left(\left(1+m\right)\times {\left(n\times N\right)}^{2}\right)$, where m is the number of surrogate data sets. So, ${TE}_{ensemble}$ is much more complex than TE. For a neuroscience experiment with channel pairs (100) × the number of surrogate data sets (1000) × stimulus conditions (4) × subjects (15), the elapsed time of ${TE}_{ensemble}$ is 240 weeks [21].

One approach to reducing the time consumption for ${TE}_{ensemble}$ is to use faster hardware devices such as FPGA [25], graphic processing unit (GPU) [21], and computer cluster [26]. ${TE}_{ensemble}$ working on GPU is one of the most effective methods to alleviate the time-consuming problem. However, ${TE}_{ensemble}$ still requires extended running time even with GPU when enormous data is involved. The experimental data in neuroscience takes about 4.8 weeks on a single GPU (NVIDIA GTX Tian) and will take longer when it searches for information transfer delay. However, the use of multiple GPUs puts forward a higher requirement on computer performance. Another approach is to use the simple estimation method. In the phase transfer entropy proposed by Lobier [27], mutual information is estimated for phase time series using a simple binning method. This approach effectively reduces the running time and can calculate the strength and the direction of interaction [28,29,30]. However, simple discretization by partitioning the observation space ignores the neighborhood relationship in continuous data, which may cause the loss of important information [31], leading to the failure of mutual information estimation in real-valued data.

Hence, to reduce the computational cost, a fast, efficient ${TE}_{ensemble}$ with a simple statistical test method is proposed here. Based on the characteristic that ${TE}_{ensemble}$ is the average value of its local transfer entropy (${te}_{ensemble}$), we use a simple *t*-test for the ${te}_{ensemble}$ of the raw data against the ${te}_{ensemble}$ from the surrogate data (one set) as the statistical test method in the novel ${TE}_{ensemble}$. Because just one set of surrogate data is used, the time consumption of the novel ${TE}_{ensemble}$ is significantly reduced. Then, we employ a widely used neural mass model (NMM) to produce neural signals through which the characteristics of the novel ${TE}_{ensemble}$ are compared with those of the traditional method. The results show that the time consumption is reduced by two or three magnitudes in the novel ${TE}_{ensemble}$. Importantly, the proposed ${TE}_{ensemble}$ robustly detects the strength and the direction of interaction, and it reaches stability with the increase in the sample size, which is slower than the traditional ${TE}_{ensemble}$. Furthermore, the novel ${TE}_{ensemble}$ can track the dynamic interaction processes between signal pairs and the effectiveness of the novel ${TE}_{ensemble}$ has also been verified in the realistic neural signals recorded from pigeons.

This paper is organized as follows: Section 2 introduces the novel ${TE}_{ensemble}$ and the NMM we used. Section 3 investigates the characteristics of the novel ${TE}_{ensemble}$ on the simulated signal pairs and the actual neural signals and compares the performance of the novel ${TE}_{ensemble}$ with that of the traditional method. Section 4 discusses the results, and Section 5 is a conclusion.

## 2. Materials and Methods

In information-theoretic framework, Shannon entropy defines the measurement of information uncertainty. For a random variable X probability distribution p(x), its Shannon entropy is:(1)$$H\left(X\right)=-\sum_{x}p\left(x\right)log_{2}p\left(x\right)$$

Shannon entropy can be extended to two random variables. For the X and Y probability distribution p(x) and p(y), the joint entropy can be defined as in Equation (2):(2)$$H\left(X,Y\right)=-\underset{x,y}{\sum }p\left(x,y\right)log_{2}p\left(x,y\right)$$

The conditional entropy in Equation (3) is the average uncertainty about x that remains when the value of y is known:(3)$$H\left(X|Y\right)=-\sum_{x,y}p\left(x,y\right)log_{2}p\left(x|y\right)$$

The mutual information between X and Y measures the reduction of one variable’s uncertainty by the knowledge of another one:(4)$$I\left(X;Y\right)=H\left(X\right)+H\left(Y\right)-H\left(X,Y\right)=-\sum_{x,y}p\left(x,y\right)log_{2}\frac{p\left(x,y\right)}{p\left(x\right)p\left(y\right)}$$

By assuming a third random variable Z, the conditional mutual information of X and Y is:(5)I(X;Y|Z)=H(X|Z)+H(Y|Z)−H(X,Y|Z)

Mutual information has been widely used in neuroscience [32]. However, the major problem is that mutual information contains no directionality. Transfer entropy, which describes the uncertainty reduction in predicting the target variable by adding the historical information of a new variable [33], is proposed to solve this deficiency. For X and Y, ${TE}_{X\to Y}$ defines the conditional mutual information between Y and $X^{-}$ (historical information of X) under $Y^{-}$ (historical information of Y):(6)$${TE}_{X\to Y}=I\left(Y;X^{-}|Y^{-}\right)=H\left(Y|Y^{-}\right)-H\left(Y|X^{-},Y^{-}\right)\;$$
Suppose (1) there is an interaction delay u between X and Y;

(2) X and Y can be approximated by a Markov process of order k and l, respectively.

With these assumptions, TE can be rewritten in a more general form as in Equation (7):(7)$$\;\;\;\;\;\;\;\;\;\;\;\;\;\;\;\;\;\;\;\;\;\;\;\;TE_{X\to Y}^{\left(k,l\right)}\left(t,u\right)=I\left(Y_{t};X_{t-u}^{\left(k\right)}|Y_{t-1}^{\left(l\right)}\right)=H\left(X_{t-u}^{\left(k\right)},Y_{t-1}^{\left(l\right)}\right)-H\left(Y_{t},X_{t-u}^{\left(k\right)},Y_{t-1}^{\left(l\right)}\right)+H\left(Y_{t},Y_{t-1}^{\left(l\right)}\right)-H\left(Y_{t-1}^{\left(l\right)}\right)\;=\sum_{y_{t},x_{t-u}^{\left(k\right)},y_{t-1}^{\left(l\right)}}p\left(y_{t},x_{t-u}^{\left(k\right)},y_{t-1}^{\left(l\right)}\right)log_{2}\frac{p\left(y_{t}|y_{t-1}^{\left(l\right)},x_{t-u}^{\left(k\right)}\right)}{p\left(y_{t}|y_{t-1}^{\left(l\right)}\right)}$$

Equation (7) can be viewed as the average of information transfer over time. Based on this, the local transfer entropy (te) is proposed [34], realizing the local or pointwise interaction (Equation (8)):(8)$$te_{X\to Y}^{\left(k,l\right)}\left(t,u\right)=i\left(Y_{t};X_{t-u}^{\left(k\right)}|Y_{t-1}^{\left(l\right)}\right)=log_{2}\frac{p\left(y_{t}|y_{t-1}^{\left(l\right)},x_{t-u}^{\left(k\right)}\right)}{p\left(y_{t}|y_{t-1}^{\left(l\right)}\right)}$$

From Equations (7) and (8), we know that the transfer entropy is the average value of the local transfer entropy.

### 2.1. Ensemble Transfer Entropy (${TE}_{ensemble}$) and Ensemble Local Transfer Entropy (${te}_{ensemble}$)

When independent repetition trials of an experimental condition meet the cyclo-stationarity, these trials are taken as an ensemble of realizations, and various probability density functions (PDFs) can be accurately estimated from the ensemble members. In this paper, we use the subscript ensemble to indicate the ensemble transfer entropy with the TE self-prediction optimality estimator:(9)$${TE}_{ensemble}\left(X\to Y,t,u\right)=I\left(Y_{t};X_{t-u}^{\left(k\right)}|Y_{t-1}^{\left(l\right)}\right)=\;\sum_{y_{t}\left(r\right),x_{t-u}^{\left(k\right)}\left(r\right),y_{t-1}^{\left(l\right)}\left(r\right)}p\left(y_{t}\left(r\right),x_{t-u}^{\left(k\right)}\left(r\right),y_{t-1}^{\left(l\right)}\left(r\right)\right)log_{2}\frac{p\left(y_{t}\left(r\right)|y_{t-1}^{\left(l\right)}\left(r\right),x_{t-u}^{\left(k\right)}\left(r\right)\right)}{p\left(y_{t}\left(r\right)|y_{t-1}^{\left(l\right)}\left(r\right)\right)}$$
where r is the number of independent repetition trials.

When the number of repetitions is sufficient to provide the necessary amount of data to estimate various PDFs in the time window $t^{\prime }\in \left[t^{-},t^{+}\right]$ reliably, the ${TE}_{ensemble}$ in $t^{\prime }$ can be estimated:(10)$$\begin{array}{c} {TE}_{ensemble}\left(X\to Y,t^{\prime },u\right)=I\left(Y_{t^{\prime }};X_{t^{\prime }-u}^{\left(k\right)}|Y_{t^{\prime }-1}^{\left(l\right)}\right)\\ \;=\sum_{y_{t^{\prime }}\left(r\right),x_{t^{\prime }-u}^{\left(k\right)}\left(r\right),y_{t^{\prime }-1}^{\left(l\right)}\left(r\right)}p\left(y_{t^{\prime }}\left(r\right),x_{t^{\prime }-u}^{\left(k\right)}\left(r\right),y_{t^{\prime }-1}^{\left(l\right)}\left(r\right)\right)log_{2}\frac{p\left(y_{t^{\prime }}\left(r\right)|y_{t^{\prime }-1}^{\left(l\right)}\left(r\right),x_{t^{\prime }-u}^{\left(k\right)}\left(r\right)\right)}{p\left(y_{t^{\prime }}\left(r\right)|y_{t^{\prime }-1}^{\left(l\right)}\left(r\right)\right)}\; \end{array}$$

With these definitions in place, we can obtain the ensemble local transfer entropy (${te}_{ensemble}$):(11)$$\;te_{ensemble}\left(X\to Y,t,u\right)=i\left(Y_{t};X_{t-u}^{\left(k\right)}|Y_{t-1}^{\left(l\right)}\right)=log_{2}\frac{p\left(y_{t}\left(r\right)|y_{t-1}^{\left(l\right)}\left(r\right),x_{t-u}^{\left(k\right)}\left(r\right)\right)}{p\left(y_{t}\left(r\right)|y_{t-1}^{\left(l\right)}\left(r\right)\right)}\;$$

### 2.2. Estimating Ensemble Transfer Entropy

A TE estimator KSG with less bias has been widely used [24]. This method is based on the nearest neighbor estimator of Kozachenko and Leonenko [35]. The distance of k-th nearest neighbor in the high-dimensional spaces are projected to the low-dimensional spaces so that the deviations caused by the different spatial scales in low-dimensional spaces are significantly reduced. In this paper, the KSG estimator is applied to ${TE}_{ensemble}$ and ${te}_{ensemble}$. Instead of searching for the nearest neighbors in the state space constructed by the individual trial, we proceed in all repetitions [36]:(12)$${TE}_{ensemble}\left(X\to Y,t,u\right)=\psi \left(k\right)+\langle \psi \left(n_{y_{t-1}^{\left(l\right)}\left(r\right)}+1\right)-\psi \left(n_{y_{t}\left(r\right)y_{t-1}^{\left(l\right)}\left(r\right)}+1\right)-\psi \left(n_{y_{t-1}^{\left(l\right)}\left(r\right)x_{t-u}^{\left(k\right)}\left(r\right)}+1\right)\rangle$$
where ψ is the Digamma function, $\psi \left(x\right)=\Gamma {\left(x\right)}^{-1}\frac{d\Gamma \left(x\right)}{dx}$; 〈.〉 means average; $n_{y_{t-1}^{\left(l\right)}\left(r\right)}$, $n_{y_{t}\left(r\right)y_{t-1}^{\left(l\right)}\left(r\right)}$, and $n_{y_{t-1}^{\left(l\right)}\left(r\right)x_{t-u}^{\left(k\right)}\left(r\right)}$ are the number of samples falling into the strip of the marginal space $y_{t-1}^{\left(l\right)}\left(r\right)$, $y_{t}\left(r\right)y_{t-1}^{\left(l\right)}\left(r\right)$, $y_{t-1}^{\left(l\right)}\left(r\right)x_{t-u}^{\left(k\right)}\left(r\right)$, respectively. The strip is defined by the distance to its k-th nearest neighbors. In general, k is 4 [23]:(13)$${te}_{ensemble}\left(X\to Y,t,u\right)=\psi \left(k\right)+\psi \left(n_{y_{t-1}^{\left(l\right)}\left(r\right)}+1\right)-\psi \left(n_{y_{t}\left(r\right)y_{t-1}^{\left(l\right)}\left(r\right)}+1\right)-\psi \left(n_{y_{t-1}^{\left(l\right)}\left(r\right)x_{t-u}^{\left(k\right)}\left(r\right)}+1\right)$$

### 2.3. Parameter Selection

The information transfer delay u between X and Y, the embedded dimension (Markov approximation order k and l) and embedded delay τ, have a significant impact on ${TE}_{ensemble}$ estimation. We use the TE self-prediction optimality estimator to obtain the transfer delay [21]. When the ${TE}_{ensemble}\left(X\to Y,t,u\right)$ is maximal, the assumed delay u is equal to the true information transfer delay δ (Equation (14)) [22]. k, l, and τ are calculated by using the Rawdgitz criterion [37]:(14)$$\delta =argmax_{u}\left(TE_{ensemlbe}\left(X\to Y,t,u\right)\right)$$

### 2.4. Surrogate Data and The Improved Statistical Test Method

${TE}_{ensemble}$ is a biased estimation with no upper bound [9], so it is necessary to generate surrogate data and construct the null hypothesis distribution to test the statistical significance of the ${TE}_{ensemble}$ metric value. In the surrogate data, it is assumed that there is no information transfer between the source variable X and the target variable Y. The commonly used method is to shuffle X, which destroys the dependence between X and Y while retaining the probability distribution of the variables [38]. Here, the source signals of each independent repetition trial are separated into two segments (Figure 1). Then, these segments are shuffled to ensure each segment is not in the same position as before. We can obtain the surrogate data ($X^{\prime }$, Y).

In the traditional ${TE}_{ensemble}$ method, at least 500 sets of surrogate data are required to generate and then ${TE}_{ensemble}$ metrics are estimated to construct the null hypothesis distribution. The null hypothesis can now be rejected or retained by comparing the ${TE}_{ensemble}$ metric value of the raw data to the null hypothesis distribution at the 1% (5%) level of significance [21].

Here, we modify the statistical test method in the traditional ${TE}_{ensemble}$. The *t*-test is a parametrical statistical significance test, which is used to test whether there is any difference in the mean values of two groups. Based on the characteristic that ${TE}_{ensemble}$ is the average value of $te_{ensemble}$ [39], a *t*-test is performed on the $te_{ensemble}$ values of the raw and the surrogate data. If the null hypothesis is rejected, it indicates that there is a significant difference between the ${TE}_{ensemble}$ of the raw data and the ${TE}_{ensemble}$ from the surrogate data.

Due to its high power, the *t*-test has been widely used to measure the difference in the mean values from two groups. In small samples, the *t*-test is valid only for data that is normally distributed [40]. However, because of the central limit theorem, the t-statistic is normally distributed with unit variance when the sample size is large, no matter what distribution the data has. Thus, the *t*-test will always be appropriate for large enough samples [41,42]. However, how large is large? The sample size relates to the difference in variance and the prevalence of extreme outliers. A large body of literatures indicate that “sufficiently large” is often less than 500 in extremely non-normally distributed data [41].

In the ensemble method, although the distributions of $te_{ensemble}$ values (which are estimated from the raw and the surrogate data, respectively) are non-normally distributed (Appendix A), the samples of $te_{ensemble}$ are often substantially larger than 500. Therefore, the *t*-test is applicable to the $te_{ensemble}$ values. We also compare the *t*-test and the Wilcoxon rank sum test (Appendix A). The results of the two methods are almost the same.

### 2.5. Neural Mass Model

Signal pairs are generated by NMM described in [43], which simulates the connectivity between multiple regions of interesting (ROIs) through long-range excitatory connections. In the NMM, the average spike density of pyramidal neurons of the presynaptic area ($Z^{X}$) affects the target region by a weight factor ω and a time delay u (Equation (15)):(15)$$u^{Y}\left(t\right)=n^{Y}\left(t\right)+\omega^{X\to Y}Z^{X}\left(t-u\right)$$
where n(t) is a Gaussian white noise. The superscripts X and Y are represented by the presynaptic and target region, respectively.

Signal pairs generated by the NMM are nonlinear and have significant β (about 20 Hz) activity. By changing the information transfer delay u and weight factor $\omega^{X\to Y}$, $\omega^{Y\to X}$, we obtain the simulated signal pairs with directional interaction.

We use the Trentool Matlab toolbox to estimate ${TE}_{ensemble}$ and ${te}_{ensemble}$ (https://github.com/trentool/TRENTOOL3_manual (accessed on 15 November 2021)). The Matlab codes of the NMM are available in ModelDB (http://modeldb.yale.edu/263637 (accessed on 1 January 2022)).

## 3. Results

In this section, we first use the simulated signal pairs to evaluate the characteristics of the novel ${TE}_{ensemble}$, and compare the performance and time consumption of the novel ${TE}_{ensemble}$ with that of the traditional method. Then, we verified the effectiveness of the novel ${TE}_{ensemble}$ in the actual neural signals.

### 3.1. Results on NMM

#### 3.1.1. The Novel ${TE}_{ensemble}$ Measures the Strength and the Direction of Interaction Robustly

The simulated signal pairs ($\omega^{X\to Y}=0,\;10,\;20,\;30,\;40,\;50,\;60,\;70$ $\omega^{Y\to X}=0,\;u=20\;\mathrm{ms}$, sampling frequency = 100, trial length = 2 s) were used to investigate whether the novel ${TE}_{ensemble}$ can reliably detect the strength and the direction of interaction between brain regions. We also explored the impact of the *t*-test significance *p* level on the results. For each $\omega^{X\to Y}$, 1000 simulated signal pairs were pooled together, then 100 pairs were drawn randomly to estimate ${TE}_{ensemble}$. This procedure was repeated 500 times and resulted in a distribution of ${TE}_{ensemble}$ values for each $\omega^{X\to Y}$. The ${TE}_{ensemble}$ metric value measured the strength of interaction. By computing the false positive rate and the sensitivity, the accurancy of the direction estimation for the novel ${TE}_{ensemble}$ was obtained. We used the same method that was introduced in [27] to calculate the false positive rate and the sensitivity. By comparing the *t*-test values for ${te}_{ensemble}$ of the raw and the surrogate data with the threshold value (the *t*-test value at the significance *p* level with the freedom degree greater than 1000), we could calculate the proportion of false positive at $\omega^{X\to Y}=0$ and the sensitivity (proportion of true positive) for $\omega^{X\to Y}\neq 0$. The threshold values were 3.0902, 2.807, 2.5756, and 2.3263, respectively, corresponding to *p* = 0.002, 0.005, 0.01, and 0.02. Then, we obtained the ${TE}_{ensemble}$ sensitivity values as a function of $\omega^{X\to Y}$ and the coupling detection threshold (CDT) for 0.8 sensitivity were computed by linear interpolation. CDT represented the smallest coupling value for which ${TE}_{ensemble}$ detected 80% of the directed interactions. Therefore, low CDT indicated that the significant interactions were detected even for weak coupling while high CDT meant that the information transfer could be detected only for solid coupling. We compared the false positive rate and the sensitivity (CDT) of the novel ${TE}_{ensemble}$ with those of the traditional ${TE}_{ensemble}$. In the traditional method, 500 sets of surrogate data were used to construct the null hypothesis distribution for each $\omega^{X\to Y}$. The false positive rate and the sensitivity (CDT) values were obtained by comparing the ${TE}_{ensemble}$ values with the null hypothesis distribution at the significance α level (α = 0.01,0.05).

Neural signals (EEG/MEG/LFP) are corrupted by both environmental and biological noise [44,45,46]. Therefore, the analysis method we apply should be robust to noise. To explore the influence of noise on the novel ${TE}_{ensemble}$, we added gaussian white noise with different energies to the simulated signals. Then, the false positive rate, sensitivity, and CDT values for coupled signal pairs with a signal-to-noise ratio (SNR) = 50, 30, 20, 10, 0, and −10 dB were calculated (the signal pairs with SNR = 50 dB were considered noise free). Finally, we compared the performance of the novel ${TE}_{ensemble}$ with that of the traditional method.

The results showed that the ${TE}_{ensemble}$ values fluctuated around 0 for $\omega^{X\to Y}=0$ and increased monotonically with $\omega^{X\to Y}$ from 0 to 70 (Figure 2a). Low, realistic noise (SNR = 50, 30, 20, 10 dB) had little effect on the ${TE}_{ensemble}$ values. The ${TE}_{ensemble}$ values decreased moderately when SNR was 0 dB, but for strong noise (SNR = −10 dB), the ${TE}_{ensemble}$ values reduced greatly. Figure 2b shows the distributions of the ${TE}_{ensemble}$ values for different $\omega^{X\to Y}$ when SNR was 20 dB. The distributions had some overlap, but the overlap areas were minimal. Therefore, different $\omega^{X\to Y}$ could be distinguished from the ${TE}_{ensemble}$ values. The novel ${TE}_{ensemble}$ in this work was consistent with the traditional method for measuring the interaction strength due to the same ${TE}_{ensemble}$ estimator they used. Above all, it suggests that the novel ${TE}_{ensemble}$ effectively measures the interaction strength even in the presence of noise.

To calculate the false positive rate and the sensitivity of the novel ${TE}_{ensemble}$, the *t*-test values were computed by performing a *t*-test for the ${te}_{ensemble}$ values, which were estimated from the raw and the surrogate data. Figure 3a shows the densities of the *t*-test values for signal pairs ($\omega^{X\to Y}=0$) with varied SNRs. They were broadly similar and normally distributed. The proportion of false positive was calculated by comparing the *t*-test values with the threshold value. It increased with the significance *p* level from 0.002 to 0.02. The false positive rate fluctuated around 0.01 when *p* was 0.02 and increased to 0.05 when *p* was 0.02 (Figure 3b). Noise had no impact on the proportion of false positive. Even if SNR was −10 dB, the false positive rate remained at a low level.

Figure 4 shows the sensitivity and the CDT values against the significance *p* levels with $\omega^{X\to Y}$ from 10 to 70. The results were twofold. First, when SNR was constant, the sensitivity values improved gradually with the increase in the significance *p* levels. The CDT value was 19.4 when *p* was 0.002 and reduced to 17.9 when *p* was 0.02. However, the decrease was minimal, and the CDT values were almost the same for varied *p* values (Figure 4e). The results indicated that the significance *p* level had a limited effect on the sensitivity and the CDT values. Second, moderate noise (SNR = 50, 30, 20, 10 dB) had little impact on the sensitivity and the CDT values. However, when SNR was 0 dB, the sensitivity values reduced, and the CDT values increased by nearly 10. Even more, the sensitivity values dramatically decreased when SNR was −10 dB and only when $\omega^{X\to Y}$ was 60, about 80% of directed interactions were detected by the novel ${TE}_{ensemble}$.

The sensitivity values of the novel ${TE}_{ensemble}$ and the traditional ${TE}_{ensemble}$ were compared in two cases (Figure 5). For case 1, the significance *p* level was 0.002 for the novel ${TE}_{ensemble}$ and the significance α level was 0.01 in the traditional method. For case 2, *p* and α were 0.02 and 0.05, respectively. In case 1, the two methods had the same false positive rate. The sensitivity values of the novel ${TE}_{ensemble}$ were somewhat lower than those of the traditional method. For SNR = 50, 30, 20, 10, 0 dB, the differences in the CDT values for the two methods were minimal, about 1.5. The difference increased to 6 when SNR was −10 dB. In case 2, the false positive rate of the traditional method was higher than that of the novel ${TE}_{ensemble}$. The two methods had the almost same sensitivity values (the CDT values) regardless of SNRs. The results indicated that the proposed ${TE}_{ensemble}$ almost had the same performance as the traditional method in detecting direction with realistic noise.

Therefore, the novel ${TE}_{ensemble}$ in this paper could robustly detect the strength and the direction of the interaction when SNR was above 0 dB, but this ability was reduced when SNR was lower than −10 dB.

Above all, the significance *p* level had a limited effect on the sensitivity and the CDT values, but it changed the proportion of false positive dramatically. Therefore, in the following, *p* = 0.002 was used as the significance level for the *t*-test in the novel ${TE}_{ensemble}$.

#### 3.1.2. Window Length and the Number of Trials Affect the Stability of the Novel ${TE}_{ensemble}$

In order to explore the interaction dynamics, a short time window is used to improve the temporal resolution. However, accurate ${TE}_{ensemble}$ estimation requires enough samples. We, therefore, investigated the effect of the window length and the number of trials on the performance of the novel ${TE}_{ensemble}$. Then, the results were compared with those of the traditional method. The samples in a time window were 100, 300, and 500, respectively. The number of trials were 50, 100, 150, and 200 with SNR = 0, 20 dB.

Figure 6a shows that when the SNR was 20 dB, the ${TE}_{ensemble}$ values clustered around 0 for $\omega^{X\to Y}$ = 0 and increased monotonically with $\omega^{X\to Y}$ from 20 to 60, regardless of the sample size in the ensemble members (Figure 6a top). The same results occurred for low SNR (Figure 6a bottom). As expected, by either increasing the window length or the number of trials, the variance of the ${TE}_{ensemble}$ values was reduced with an increased sample size.

Figure 6b shows the sensitivity of the novel ${TE}_{ensemble}$ with varied window lengths and the number of trials when SNR was 0, 20 dB. The longer the window length and the more trials used, the higher the sensitivity ${TE}_{ensemble}$ obtained. Under the same conditions, the sensitivity for SNR = 20 dB is higher than that of SNR = 0 dB.

To quantify the sensitivity, we calculated the CDT values for a varied amount of simple sizes with SNR = 50, 20, 0 dB and compared the CDT values of the novel ${TE}_{ensemble}$ with those of the traditional method. The results showed that the CDT values for the two methods gradually reduced with the increase in the sample size, regardless of the SNR values, and finally reached the same stable state (Figure 6c). For SNR = 50, 20 dB, the CDT values in the traditional ${TE}_{ensemble}$ reached stability when the sample size was 15,000 while it was 20,000 in the novel ${TE}_{ensemble}$ The same results were obtained when SNR was 0 dB. The traditional method reached stability with 75,000 samples, which was faster than that of the novel ${TE}_{ensemble}$.

Finally, we calculated the false positive rate for different sample sizes. Figure 6d shows that the false positive rate fluctuated between 0 and 0.02 for the two methods regardless of the samples and noise. There were several values above 0.02 in the traditional method. The false positive rate of the novel ${TE}_{ensemble}$ seems to be more stable. Therefore, the window length and the number of trials affect the stability of the novel ${TE}_{ensemble}$. With the increase in sample size, the variance of the ${TE}_{ensemble}$ values was reduced and the sensitivity of the novel ${TE}_{ensemble}$ reached its steady state, but it was slower than that of the traditional method.

#### 3.1.3. The Novel ${TE}_{ensemble}$ Requires Less Computation Time to Track the Dynamic Interaction Process

To evaluate the ability and computation time of the novel ${TE}_{ensemble}$ in this work in tracking the dynamic interaction process, signal pairs were generated with varying coupling strength (u = 20 ms, trial length = 60 s, number of trials = 100, sampling frequency = 100 Hz). $\omega^{X\to Y}$ was on/off boxcar and $\omega^{Y\to X}$ varied with the absolute value of a sinusoid (Figure 7b). Then, we used a scanning approach to reconstruct ${TE}_{ensemble}$ values and the corresponding interaction delay u. We scanned assumed delays in the interval u=[0,30] ms with 10 ms steps and a window length ∆t=2s was used for the novel and the traditional ${TE}_{ensemble}$. Because they used same estimation method, the ${TE}_{ensemble}$ values for both could track the dynamic process of $\omega^{X\to Y}$ and $\omega^{Y\to X}$, and bidirectional coupling had no effect on each other (Figure 7c,d).

The time consumptions were compared for the two methods on GPU and CPU, respectively. For the above bidirectional interaction, the computation time of the traditional method on CPU (CPU-Intel(R) Core (TM) i5-10210U) was 1,237,322.12 s (500 surrogate data sets were used), and the running time on GPU was 37,494.6 s while the time consumption of the novel ${TE}_{ensemble}$ was 9634.99 s on CPU, and just 291.97 s (4.87 min) on GPU (Figure 7e). The time consumption on GPU was calculated to be 33 times faster than that on CPU. Due to only one set of surrogate data used, the time consumption of the novel ${TE}_{ensemble}$ was significantly lower than that of the traditional method, and the novel ${TE}_{ensemble}$ solved the problem of computational complexity fundamentally.

### 3.2. Applying the Novel ${TE}_{ensemble}$ on the Actual Neural Signals

To explore the applicability of ${TE}_{ensemble}$ in this paper for the actual neural signals, the local field potentials (LFPs) of hippocampal (Hp) and nidopallium caudolaterale (NCL) were recorded on two pigeons (P087, P089) when they were performing a goal-directed decision-making task. The pigeons were trained to start from the waiting area, pass through the straight area, and turn left, forward, or right as the goal location light instructed. If the pigeons choose the correct direction, they could obtain food (Figure 8). Details of the experimental process are described in [47].

Data processing was performed using custom scripts written in MATLAB. First, we removed the trials containing strong motion artifact. The adaptive common average reference was used for all channels of the remaining trials [46] and 50 Hz line noise was suppressed using an adaptive notch filter. Then, the signals were resampled to 1000 Hz and the slow gamma frequency band (40–60 Hz) of LFPs recorded from Hp and NCL were extracted for analysis. Finally, from the waiting to turning period, a total of 4-s signals were used. The number of independent repetition trials for each pigeon was 500.

We used the novel ${TE}_{ensemble}$ that was introduced in this work to calculate the dynamic interaction between Hp and NCL of the pigeon in the goal-directed decision-making tasks. A time window of ∆t=200 ms was used. For each non-overlapping time window, we scanned the assumed interaction delays in the interval u=[10, 50] ms with 5 ms steps. In total, 100 trials were used by randomly drawing from 500 trials for ${TE}_{ensemble}$ estimation. This procedure was repeated 10 times. The results of the novel ${TE}_{ensemble}$ were consistent with Zhao’s, using the functional network and partial directional coherence [47]. The information flowed from Hp to NCL in the turning period with a 30 ms time delay. The novel ${TE}_{ensemble}$ reported the dynamic interaction process in detail and found that the interaction between the two brain regions reached the maximum at about 1 s after the animal entered the turning area (Figure 9). This may be related to the fact that the animal saw the light stimulation after entering the turning area, and then transferred the spatial location information formed in Hp to NCL. Therefore, the novel ${TE}_{ensemble}$ in this paper is suitable for analyzing the dynamic interaction process between the actual neural signals.

## 4. Discussion

To reduce the time consumption of the ensemble transfer entropy (${TE}_{ensemble}$) and explore the dynamic interaction process in neuroscience, we proposed a fast, efficient ${TE}_{ensemble}$ in which we modified the traditional statistical method. A *t*-test for the ${te}_{ensemble}$ values that were estimated from the raw and the surrogate data was performed to test whether there was a significant difference in their mean values-${TE}_{ensemble}$. Because just one set of surrogate data was used, the time consumption of the novel ${TE}_{ensemble}$ was significantly reduced. To validate the improved efficiency, the coupled signal pairs generated by a neural mass model were used. First, the novel ${TE}_{ensemble}$ in this paper robustly detected the strength and the direction of the interaction between signal pairs with moderate noises (SNR was above 0 dB) and its performance decreased dramatically when SNR was −10 dB. It yielded almost the same false positive rate and sensitivity as those of the traditional ${TE}_{ensemble}$. Second, with the increase in the window length and the number of trials, the novel ${TE}_{ensemble}$ reached its stable state, but it was slower than that the traditional method. Third, the novel ${TE}_{ensemble}$ could accurately track the dynamic interaction process and its computation time was reduced by two to three orders of magnitude compared with the traditional method. Finally, the applicability of the novel ${TE}_{ensemble}$ in the realistic neural signals was verified on the LFP signals of Hp and NCL when pigeons performed goal-directed decision-making tasks. Therefore, the novel ${TE}_{ensemble}$ in this paper may be a suitable way to investigate the dynamic interaction process between brain regions.

${TE}_{ensemble}$ is a biased estimation and does not have a meaningful upper bound [9], so it is necessary to construct the null hypothesis distribution to test the statistical significance of ${TE}_{ensemble}$, which is an essential part in ${TE}_{ensemble}$ [21]. In TE, the null hypothesis distribution can be constructed by one set of surrogate data (n trials) [9]. However, in the ensemble method, only a single metric value can be obtained from n trials [20], so the null hypothesis distribution is built by multiple sets (usually more than 500) of surrogate data, which dramatically increases the amount of calculation [21]. In this paper, we introduced a simple statistical method in the novel ${TE}_{ensemble}$ by performing a *t*-test for ${te}_{ensemble}$ of the raw and the surrogate data (one set). Because just one set of surrogate data was needed, the computation time was reduced significantly and the computational complexity in ${TE}_{ensemble}$ was fundamentally solved. However, there is still a large amount of calculation in the novel ${TE}_{ensemble}$ with the KSG estimator. For the construction of a multi-brain dynamic interaction network, one workaround is that the novel ${TE}_{ensemble}$ runs on GPU. Another way is to use the TE estimator with a small amount of calculation, for instance, the symbolic version of TE based on ordinal pattern symbolization, kernel-based transfer entropy, and the transfer entropy rate through Lempel–Ziv complexity. The next step is to generalize these estimators to the ensemble method and compare their performance in an ensemble of realizations.

One may wonder whether the service conditions of the *t*-test are met in the novel ${TE}_{ensemble}$. In fact, the ${te}_{ensemble}$ of the raw data and the surrogate data are not normally distributed (Appendix A). However, based on the central limit theorem, the *t*-test is always appropriate for large enough samples, regardless of the distribution of the data [41,42,48]. In the ensemble method, the sample size of ${te}_{ensemble}$ is lager (generally more than 5000). So, it is stable to use the *t*-test to detect whether there is any significant difference between ${TE}_{ensemble}$ of the raw data and ${TE}_{ensemble}$ from the surrogate data. Meanwhile, we compared the false positive rate and the sensitivity of the novel ${TE}_{ensemble}$ with the *t*-test and Wilcoxon rank-sum test. The two methods obtained the same false positive rate and CDT values (Appendix A). Therefore, in this paper, it is possible to use a *t*-test of ${te}_{ensemble}$ values from the raw data against ${te}_{ensemble}$ values from the surrogate data to detect the significant difference between the ${TE}_{ensemble}$ values of the raw and the surrogate data.

In the novel ${TE}_{ensemble}$, we chose *p* = 0.002 as the significance level for the *t*-test. The false positive rate fluctuated around 0.01 when *p* was 0.002 and increased to 0.05 when *p* was 0.02. Someone may doubt whether the novel ${TE}_{ensemble}$ is able to control the false positive rate at the desired level and they believe there is just a 1% chance of their result being a false alarm when *p* is 0.01. However, the false positive rate is not only related to *p* value but also intimately connected to the sample size. When the sample size is large, the sensitivity of the statistical test method is very high. The result is positive even for the two groups with a small difference [49]. In the ensemble method, the samples of ${te}_{ensemble}$ are usually larger than 5000. If we expect the false positive rate to be 0.01, we should reduce the *p* value instead of *p* = 0.01. Meanwhile, the results in Section 3.1.1 also confirmed this conclusion. Only when *p* is 0.002, the false positive rate is around 0.01, and it is 0.03 instead of 0.01 when *p* is 0.01. So, the *p*-value does not measure the probability that the studied hypothesis is true. It reflects our level of tolerance for the false positive rate [50,51].

One of the major challenges in brain science is that the neural signals are corrupted by technical noises (power line interference, impedance fluctuation, motion artifacts, etc.) and biological artifacts (volume conductor, eye movement, eyeblink, muscular, etc.) [52,53,54,55]. The presence of noise can mask the features of the neural signals and affect the analysis of the interactions between brain regions. Various methods have been proposed to eliminate noise. For instance, the elimination of noise at the source by standardizing the experimental operation [45], reduction in the power line interference by the adaptive notch filter [56], removal of muscle artifact by ensemble empirical mode decomposition and multiset canonical correlation analysis [57], and so on. However, some filters that are obtained by convolution of the input with their impulse responses may blur the temporal or causal relations between signal and external events [45]. We should be cautions when using them. If signals are recorded on multiple channels, spatial filters may be applied to remove noise [46]. However, some noises are sufficiently complex so that we cannot disentangle them completely from neural processing, which we really need. Therefore, the analysis methods we use should be robust to noise. In Section 3.1.1, we investigated the robustness of the novel ${TE}_{ensemble}$ to noise. The results show that the novel ${TE}_{ensemble}$ can measure the strength and the direction of interaction robustly when the SNR is above 0 dB. Therefore, the novel ${TE}_{ensemble}$ we propose is valid in the presence of moderate noises.

${TE}_{ensemble}$ based on the dependent repetition trials detects interaction within a short time window. It has high temporal resolution and is suitable for analyzing the dynamic interaction process between neural signals [20]. However, the selection of the time window length needs to pay attention to the following points: First, we used a scanning method to obtain the interaction time delay u. However, the time delay can only be estimated accurately when the sample size is greater than 10,000 [21]. Therefore, in order to estimate the time delay accurately in collapse trials, the length of the time window should be selected to ensure that there are enough samples in the ensemble members. Second, in the novel ${TE}_{ensemble}$, to obtain an accurate estimation of future information, enough historical information should be involved. We used the Ragwitz criterion to calculate the embedded window (embedded dimension k,l * embedded delay τ), which includes the past information of the source and the target signals and has the ability to predict the future of the target signal [37]. So, the larger the embedded window, the longer the time window that should be picked. Finally, the selection of the time window length is limited by the number of independent repetition trials. Based on the analysis in Section 3.1.2, the performance of the novel ${TE}_{ensemble}$ is affected by the sample size. The larger the sample size, the better the stability of the novel ${TE}_{ensemble}$. The sensitivity and the CDT values reached stability when the sample size was more than 20,000 with moderate noise. Therefore, we can select a small window to improve the temporal resolution with more independent repetitions. When the number of independent repetitions is less, a large time window should be used to ensure the stability of the performance.

In TE, the KSG estimator requires the signal to be stationary to obtain accurate results [32]. However, this is difficult to realize in neuroscience and most neural signals are non-stationary. ${TE}_{ensemble}$ solves this problem by estimating using independent repetitions in which equivalent events (or equivalent brain activity) occur periodically. The neural signals recorded from these trials are assumed to be cyclo-stationary [58]. In general, independent repetitions meet this hypothesis [21]. However, the brain activity in the learning tasks changes gradually and its neural signals are not cyclo-stationary, so in this case, caution should be exercised when using the novel ${TE}_{SPO}$.

Local transfer entropy is a time-varying version of TE, which was proposed by Lizier to realize the local or pointwise interaction estimation, and TE can be expressed as an average of local transfer entropy [39]. In recent years, local transfer entropy has been used in neuroscience [59]. Ramón demonstrated that the local mutual information is suitable for measuring the dynamics of cross-frequency coupling in brain electrophysiological signals [32]. Local transfer entropy has been applied to explore the dynamic coupling of the phase–amplitude during seizures [59]. However, Sezen questions the local causality measure of local transfer entropy because the causal nature does not necessarily remain in each part [60]. In addition, there is high-frequency leakage into the local transfer entropy, and it is difficult to explain this phenomenon theoretically. The use of local transfer entropy to measure dynamic interaction needs further research. In this paper, local transfer entropy is not directly used to investigate the dynamic interaction between signal pairs. We used the characteristic that TE is the average value of the local transfer entropy for the statistical analysis in the novel ${TE}_{ensemble}$.

Research in rodents and avian has shown that goal-directed behavior involves multiple brain regions. Among them, Hp and PFC/NCL play important roles. Hp participates in goal-directed behavior by recognizing key locations in space [61]. PFC/NCL is involved in weighing conflicting, then making a decision [62]. Hp and PFC/NCL have very close functional interactions that contribute to goal-directed behavior’s successful execution [63]. In a previous work, we investigated the interaction between Hp and NCL of pigeons in a goal-directed task using the local function network and partial directional coherence. The results show that during the turning area, the functional interaction of the Hp-NCL increases significantly, and the information flows from Hp to NCL [48], which was also detected using the novel ${TE}_{ensemble}$. However, the whole decision-making period is as long as 2 s, in which the dynamics of Hp–NCL interaction is unknown. The novel ${TE}_{ensemble}$ solves this problem and it was found that the ${TE}_{ensemble}$ metric value reaches the maximum at about 1 s after the animal entered the turning area. This may be due to the spatial location information forming in Hp when the animals saw the light stimulation in the turning area, and then it being transferred from Hp to NCL for decision-making. The results show that the novel ${TE}_{ensemble}$ is suitable for investigating the dynamic interaction between the actual neural signals.

## 5. Conclusions

We introduced a fast and efficient ensemble transfer entropy (${TE}_{ensemble}$) to detect the dynamic interaction process between neural signals. It uses a *t*-test for the ensemble local transfer entropy (${te}_{ensemble}$) from the raw data against the ${te}_{ensemble}$ from the surrogate data as the statistical method in the novel ${TE}_{ensemble}$. Due to just one set of surrogate data being used, the time consumption is significantly reduced. We compared the performance of the novel ${TE}_{ensemble}$ with that of the traditional ${TE}_{ensemble}$. The novel ${TE}_{ensemble}$ exhibited many characteristics. First, the time consumption was reduced by two or three magnitudes. Second, the novel ${TE}_{ensemble}$ reliably measures the strength and the direction of the interaction in the presence of moderate noise. Compared with the traditional ${TE}_{ensemble}$, the novel ${TE}_{ensemble}$ was slower to reach its steady state with the increase in the sample size. Third, the novel ${TE}_{ensemble}$ could track the dynamic interaction process accurately. Finally, the novel ${TE}_{ensemble}$ was applicable to the actual neural signals. Taken together, the novel ${TE}_{ensemble}$ may be a suitable method for quantifying the dynamic interactions in neuroscience.

## Figures and Tables

**Figure 1 entropy-24-01118-f001:**
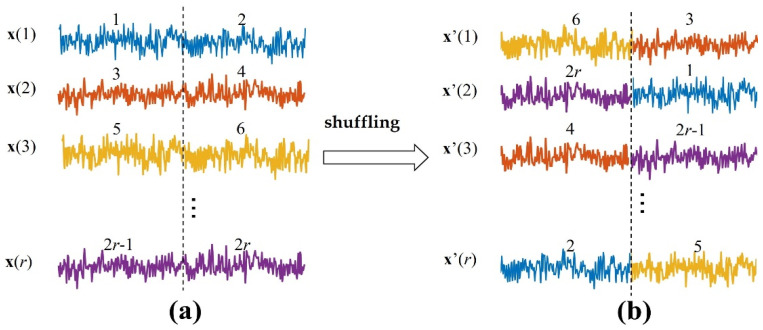
Generating the surrogate data. (**a**) x(1), x(2),⋯,x(r) are the source signals of independent repetition trials. Each trial is separated into two segments, these segments are shuffled in the ensemble members to ensure each segment is not in the same position as before, and then the surrogate data $x^{\prime }\left(1\right),\;x^{\prime }\left(2\right),\;\cdots ,x^{\prime }\left(r\right)$ in (**b**) are generated.

**Figure 2 entropy-24-01118-f002:**
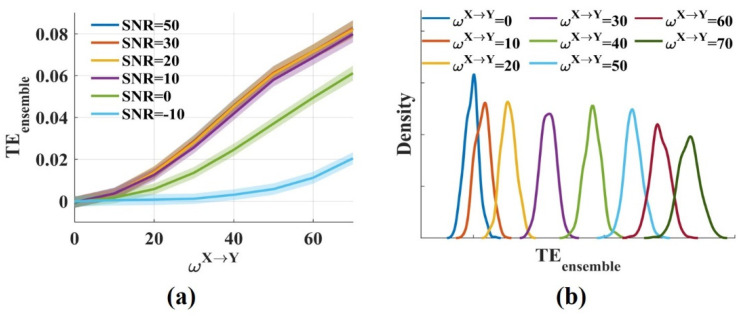
The novel ${TE}_{ensemble}$ measures the interaction strength robustly. (**a**) ${TE}_{ensemble}$ values fluctuated around 0 when $\omega^{X\to Y}$ was 0 and increased monotonically with $\omega^{X\to Y}$ from 0 to 70. The solid lines and shaded areas represent the mean and the variance of the ${TE}_{ensemble}$ values, respectively; (**b**) The densities of the ${TE}_{ensemble}$ values for varied $\omega^{X\to Y}$(SNR = 20 dB) were computed. The distributions had some overlap, but the overlap areas were minimal.

**Figure 3 entropy-24-01118-f003:**
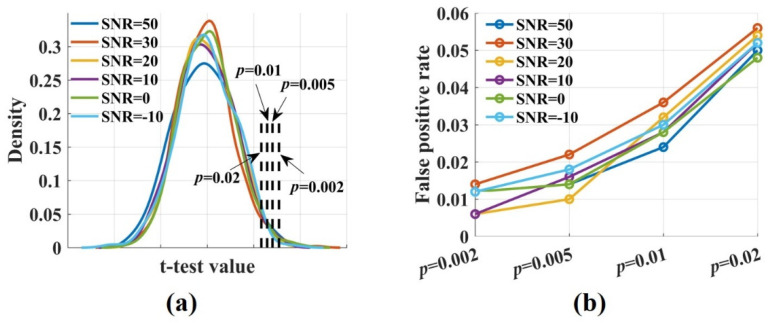
The densities of the *t*-test values and the false positive rate at different significance *p* levels. (**a**) The densities of the *t*-test values with $\omega^{X\to Y}=0$, SNR = 50, 30, 20, 10, 0, −10 dB were broadly similar and normally distributed; (**b**) The false positive rate increased with the significance level-*p* from 0.002 to 0.02. It fluctuated around 0.01 when *p* was 0.002 and increased to 0.05 when *p* was 0.02. Noise had no impact on the proportion of false positive.

**Figure 4 entropy-24-01118-f004:**
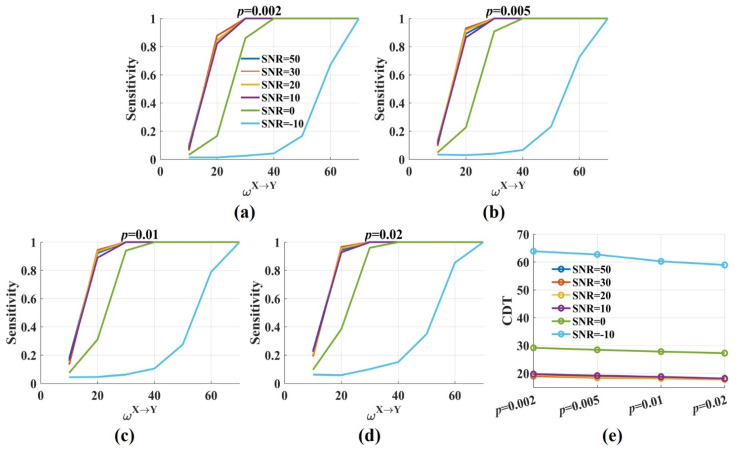
The sensitivity and the CDT values of the novel ${TE}_{ensemble}$ with varied significance *p* levels. (**a**–**d**) The sensitivity values of ${TE}_{ensemble}$ were plotted against the weight factor $\omega^{X\to Y}$ for different SNRs with the significance *p* level = 0.002, 0.005, 0.01, 0.02. The sensitivity values improved with *p* from 0.002 to 0.02. Moderate noise had a limited effect on the sensitivity. However, the sensitivity values were reduced for low SNR; (**e**) The CDT values (sensitivity = 0.8) were plotted against the significance *p* levels with varied SNRs. Moderate noises and *p* values had a limited effect on the CDT values, and low SNR increased them.

**Figure 5 entropy-24-01118-f005:**
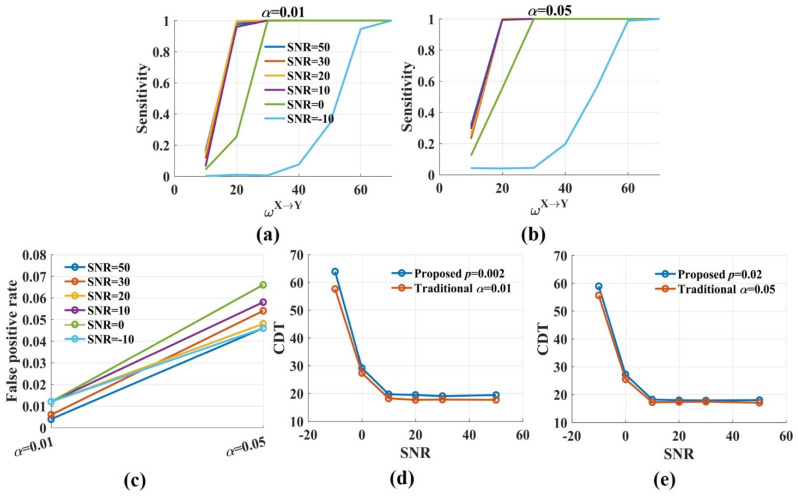
The sensitivity and the CDT values of the traditional ${TE}_{ensemble}$ when the significance α level was 0.05 and 0.01, respectively. (**a**,**b**) The sensitivity values in the traditional method were increased with α from 0.01 to 0.05. Moderate noises had a limited effect on sensitivity, but the sensitivity values dramatically reduced when SNR was −10 dB; (**c**) The false positive rate in the traditional ${TE}_{ensemble}$ fluctuated around 0.01 when the significance α level was 0.01 and increased to 0.05 or higher when α was 0.05; (**d**,**e**) The CDT values of the novel ${TE}_{ensemble}$ were compared with those of the traditional method for varied SNRs. For SNR = 50, 30, 20, 10, 0 dB, the differences in the CDT values for the two methods were tiny. The differences increased when SNR was −10 dB.

**Figure 6 entropy-24-01118-f006:**
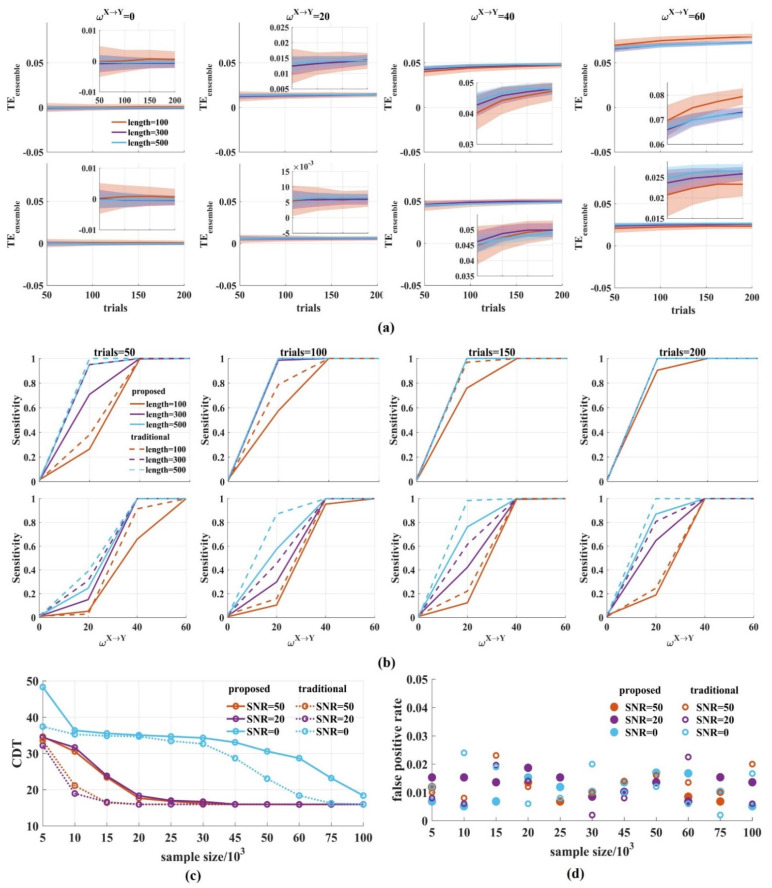
The performance of the novel ${TE}_{ensemble}$ with varied window lengths and number of trials. (**a**) We calculated the ${TE}_{ensemble}$ values for varied window lengths (100, 300, 500) and number of trials (50, 100, 150, 200) (top: SNR = 20 dB; bottom: SNR = 0 dB). In the upper (or bottom) right-hand corner is a detailed drawing. The ${TE}_{ensemble}$ values clustered around 0 for $\omega^{X\to Y}$ = 0 and increased monotonically for $\omega^{X\to Y}$ from 20 to 60, regardless of the SNR values. The variance of the ${TE}_{ensemble}$ values was reduced with the increase in the sample size. The solid lines and shaded areas represent the mean and the variance of the ${TE}_{ensemble}$ values, which were calculated 500 times by drawing a certain number of pairs from 1000 signal pairs; (**b**) The sensitivity of the novel ${TE}_{ensemble}$ was calculated. The solid lines are the sensitivity values of the novel ${TE}_{ensemble}$ and the dotted lines are those of the traditional method. With the increase in the sample size, the sensitivity improved (top: SNR = 20 dB; bottom: SNR = 0 dB); (**c**) The CDT values against the sample size with SNR = 50, 20, 0 dB. With the increase in samples, the CDT values reached their stable state in the novel ${TE}_{ensemble}$, which was slower than that of the traditional ${TE}_{ensemble}$; (**d**) We obtained the false positive rates for different sample sizes with SNR = 50, 20, 0 dB. They fluctuated between 0 and 0.02 for the two methods regardless of the samples and noise. There were several values above 0.02 in the traditional method.

**Figure 7 entropy-24-01118-f007:**
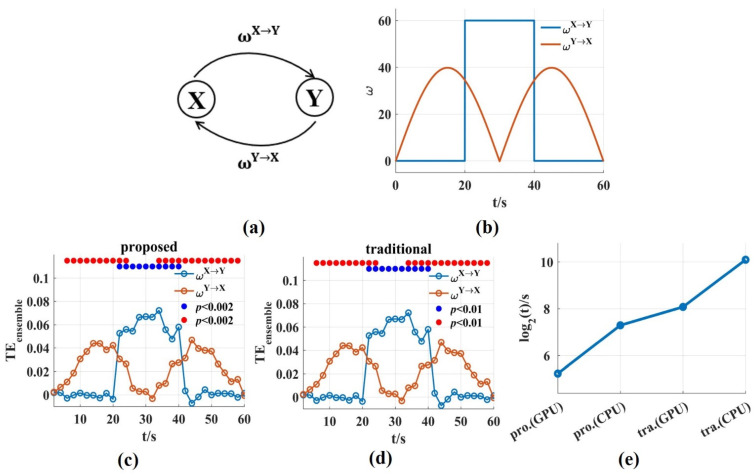
The ability and the time consumption of the novel ${TE}_{ensemble}$ in tracking the dynamic interaction process. (**a**,**b**) Signal pairs were generated with varying coupling strength. $\omega^{X\to Y}$ is the on/off boxcar and $\omega^{Y\to X}$ varies with the absolute value of a sinusoid; (**c**,**d**) The ${TE}_{ensemble}$ values for the novel and the traditional method could track the dynamic process of $\omega^{X\to Y}$ and $\omega^{Y\to X}$. The solid blue and red circles indicated the significant interaction. (**e**) The time consumptions of the novel and the traditional ${TE}_{ensemble}$ on GPU and CPU, respectively. The time consumption was reduced by about two or three orders of magnitude compared with that of the traditional method. The pro. and tra. are the abbreviations of proposed and traditional, respectively.

**Figure 8 entropy-24-01118-f008:**
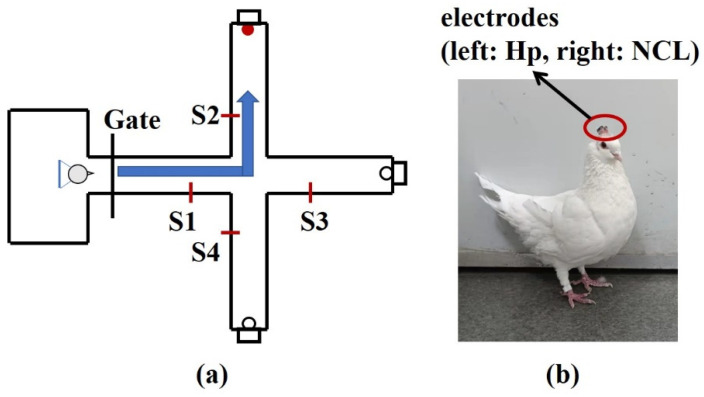
Diagrammatic sketch of a plus-maze and pigeon with implanted arrays. (**a**) The pigeons were trained to start from the waiting area, pass through the straight area, and turn left, forward, or right as the goal location light instructed. After a reward was consumed, they returned to the waiting area. S1, S2, S3, S4 are infrared sensors. (**b**) The microelectrode arrays were implanted at Hp and NCL.

**Figure 9 entropy-24-01118-f009:**
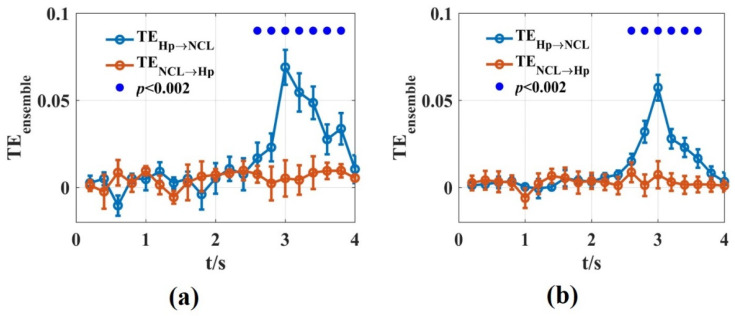
The ${TE}_{ensemble}$ values of the Hp and NCL brain regions in pigeons when they were preforming the goal-directed decision-making tasks. (**a**,**b**) ${TE}_{ensemble}\;\mathrm{values}$ were calculated using the LFPs recorded from Hp and NCL for pigeon P087 (**a**) and P089 (**b**). The blue lines are the ${TE}_{ensemble}$ values from Hp to NCL and the red lines are the opposite. The solid blue circles represented the significant interaction from Hp to NCL when the significance *p* level was 0.002.

## Data Availability

The Trentool Matlab toolbox for ${TE}_{SPO}$ estimation is open source and can be download at https://github.com/trentool/TRENTOOL3_manual (accessed on 15 November 2021). Matlab codes of the neural mass model are available in ModelDB (A neural mass model for critical assessment of brain connectivity; Ursino et al., 2020) at http://modeldb.yale.edu/263637 (accessed on 1 January 2022). The actual neural data presented in this study are available on request from the corresponding author.

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
