# Peer review of "A Fast and Efficient Ensemble Transfer Entropy and Applications in Neural Signals"

_entropy, 2022, doi:10.3390/e24081118_

Round 1

Reviewer 1 Report

The authors present a novel method for assessing the statistical significance of estimate transfer entropy (TE) values. Existing approaches to TE estimation and the associated statistical tests often use permutation tests against surrogate data, which is computationally expensive. The proposed method instead uses a t-test of local TE values against local TE values from surrogate data.

The problem, the authors are addressing is an important one and computational run times often hinder the application of information-theoretic methods in data analysis. However, the proposed method (presumably) uses a parametric test, which is a parametric test and up to date an analytical surrogate distribution is an open research question for the nearest-neighbor estimator used here (see, for example, Bossomaier, 2016 An Introduction to Transfer Entropy, Springer, p.90-91). Hence, it remains doubtful if the proposed solution leads to reliable assessments of the statistical significance of estimated TE values (see also more detailed comments to Methods and Results Sections).

# Abstract

What is meant by "high resolution". Per-se, the TE is not  measure with high temporal resolution, but typically an average measure over time.
Also in TE_SPO, the SPO refers to self prediction optimality (Wibral, 2013). This is not the ensemble TE.

# Introduction

Again, TE_SPO refers to self prediction optimality as introduced in Wibral (2013) [Ref. 13]. The ensemble estimation for the TE_SPO was introduced in Wollstadt (2014) [Ref. 21]. Please correct l. 55 and the following sentences: Gomez-Herrero proposed an estimator for TE, while Wibral (2013) introduced the reconstruction of information-transfer delays and the TE_SPO estimator, and Wollstadt (2014) introduces the estimation from ensembles of copies of processes. Please correct the introduction of the TE_SPO estimator as well as the ensemble method for its estimation (see also next paragraph).

Please take care to use the correct names of authors:
Germán -> Gómez-Herrero
Lizzer -> Lizier

# Methods
In Lines 151-152, the authors introduce the subscript SPO to indicate the ensemble method. However, this is in contradiction to the literature, where TE_SPO refers to self prediction optimality as introduced in Wibral (2013). I would recommend to change the notation of TE_SPO and choose a different subscript to indicate the ensemble estimation for TE.

In Section 2.4, the newly introduced generation of the surrogate data should be described in more detail (especially, the content in lines 199-202). Currently it is hard to follow the explanation. Is the proposed t-test a parametric test? If so, a discussion whether such a test is applicable to TE data would be required. Currently, most applications and toolboxes using TE, use a non-parametric permutation test, because estimated TE values do not fulfill the assumptions of parametric tests. Instead, permutation tests are used as in the traditional TE estimation method in the TRENTOOL toolbox. Hence, a theoretical justification of the t-test applied here would be necessary because in my understanding, a t-test is not applicable to (local) TE values.

# Results

The results show that the false positive rate is higher than the desired critical value set for p (e.g., Fig. 3B). This would mean, the method is not able to control the false positive rate at the desired level indicated by p. In other words, for p=0.01, one would expect a false positive rate of 0.01, but here it is 0.03. The sames is true for all other values investigated for p. Doesn't this mean, the method is not able to control the false positive rate at the desired level?

Figure 6: It would be helpful to see the sensitivity of the old compared to the new method (Fig. 6b).

Typos:
Blanks are missing before all citations.
l.46: Reformulate sentence.
l.55: I believe, this should be Gomez-Herrero?
l.114: remove the "and"
l.147: Replace "Lizzer" bz "Lizier"
l.148: Should this line be a subsection?
l.260: Check commas and white space in list of SNR values
l.572: Remove blank between TE_SPO and punctuation mark

Reviewer 2 Report

Recommendation: Accept in current form.

The manuscript is an efficient ensemble transfer entropy (?????) to detect the dynamic interaction process between neural signals. It uses only a t-test.

Multiple signals are extrapolated for one neural network. 

Hopefully, the method can be extrapolated further for multiple algorithmc signals from two or more neural networks.  

The work is ready for publication in Entropy

Reviewer 3 Report

July 11, 2022

I find the idea presented in the paper interesting and promising but I have the following comments and concerns:  
A. A lack of a reference to noise-related problems in the transmission channels, in particular in the biological systems, like in the following papers:
- Kanitscheider, I.; Coen-Cagli, R.; Pouget, A. Origin of information- limiting noise correlations. Proc. Natl. Acad. Sci. USA 2015, 112, e6973–e6982..
- Pregowska, A. Signal Fluctuations and the Information Transmission Rates in Binary Communication Channels. Entropy 2021, 23, 92.
- Zhang, M.L.; Qu, H.; Xie, X.R.; Kurths, J. Supervised learning in spiking, neural networks with noise-threshold. Neurocomputing 2017, 219, 333–349
- Wang, Y.; Xu, H.; Li, D. et al. Performance analysis of an adaptive optics system for free-space optics communication through atmospheric turbulence. Sci. Rep. 2018, 8, 1124.
I recommend adding (including) into the paper a Paragraph, dealing with this issue among others on the above references.
B. The language should be carefully revised.

The idea presented and developed in the paper seems to be interesting and the results obtained are promising, but due to the above concerns, I would not recommend this paper for publication until the above comments/questions will be carefully addressed. At this moment I would recommend at least a Major Revision.

Round 2

Reviewer 1 Report

I thank the authors for the careful consideration of the reviewers' comments. To my feeling the additional explanations, in particular, on the method's background and on the generation and testing of surrogate data.

I propose to publish the paper after checking for typos and spelling errors (see below for some minor corrections). Also, I suggest to carefully check the references as these seemed to be incorrect in some cases (see below).

l. 54: I think [16] is not the correct reference here?
l. 56: Again, [19] may not be the correct reference?
l. 438: change "(figure 7(c-d))" to "(Figure 7(c-d))"
l. 529: change "with the [48]KSG estimator" to "with the KSG estimator [48]", also, [48] may not be the correct reference.

Reviewer 3 Report

The Authors took my comments into account and I recommend this paper for publication.